# Cluster-Phys: Facial Clues Clustering Towards Efficient Remote Physiological Measurement

Wei Qian*
School of Computer Science and Information Engineering, School of Artificial Intelligence, Hefei University of Technology
qianwei.hfut@gmail.com

Kun Li*
CCAI, Zhejiang University
kunli.hfut@gmail.com

Dan Guo†
Hefei University of Technology, Institute of Artificial Intelligence (IAI), Hefei Comprehensive National Science Center
guodan@hfut.edu.cn

Bin Hu
Gansu Provincial Key Laboratory of Wearable Computing, School of Information Science and Engineering, Lanzhou University
bh@lzu.edu.cn

Meng Wang†
Hefei University of Technology, Institute of Artificial Intelligence (IAI), Hefei Comprehensive National Science Center
eric.mengwang@gmail.com

## Abstract

Remote photoplethysmography (rPPG) measurement aims to estimate physiological signals by analyzing subtle skin color changes induced by heartbeats in facial videos. Existing methods primarily rely on the fundamental video frame features or vanilla facial ROI (region of interest) features. Recognizing the varying light absorption and reactions of different facial regions over time, we adopt a new perspective to conduct a more fine-grained exploration of the key clues present in different facial regions within each frame and across temporal frames. Concretely, we propose a novel clustering-driven remote physiological measurement framework called Cluster-Phys, which employs a facial ROI prototypical clustering module to adaptively cluster the representative facial ROI features as facial prototypes and then update facial prototypes with highly semantic correlated base ROI features. In this way, our approach can mine facial clues from a more compact and informative prototype level rather than the conventional video/ROI level. Furthermore, we also propose a spatial-temporal prototype interaction module to learn facial prototype correlation from both spatial (across prototypes) and temporal (within prototype) perspectives. Extensive experiments are conducted on both intra-dataset and cross-dataset tests. The results show that our Cluster-Phys achieves significant performance improvement with less computation consumption. The source code will be available at https://github.com/VUT-HFUT/ClusterPhys.

*Wei Qian and Kun Li contributed equally to this research.
†Corresponding authors

## CCS Concepts

• **Computing methodologies → Artificial intelligence**.

## Keywords

Remote photoplethysmography, prototypical clustering, physiological measurement, facial videos

**ACM Reference Format:**
Wei Qian, Kun Li, Dan Guo, Bin Hu, and Meng Wang. 2024. Cluster-Phys: Facial Clues Clustering Towards Efficient Remote Physiological Measurement. In *Proceedings of the 32nd ACM International Conference on Multimedia (MM '24), October 28-November 1, 2024, Melbourne, VIC, Australia.* ACM, New York, NY, USA, 10 pages. https://doi.org/10.1145/3664647.3680670

## 1 Introduction

Due to the periodic nature of the human heartbeat [20, 21, 54], blood volume undergoes corresponding changes, manifesting in variations in the skin's light absorption rate [21, 47]. Although the skin color change is imperceptible to the naked eye, this subtle change in skin color can be recorded by an ordinary camera. Building on this premise, remote photoplethysmography (rPPG) technology [20, 21, 35, 36, 58] has been developed for physiological signal measurements like estimating heart rate (HR), heart rate variability (HRV), and respiration frequency (RF). With the convenience and non-intrusion nature, vision-based rPPG-based physiological measurement has become a research hotspot and has been widely applied in driver monitoring [10], atrial fibrillation screening [25], and face anti-spoofing [13, 56].

Early studies primarily analyze the subtle skin color changes with traditional signal processing algorithms, such as blind source separation [14, 33] and color space transformation [3, 49]. However, they heavily rely on prior knowledge or assumptions such as skin reflection model [3, 49] or linear combination assumption [33, 34], only applicable in well-controlled laboratory environments. Then, data-driven methods [26, 27, 43, 58, 60] dominate this domain. 2D-CNN [23, 32, 41] and 3D-CNN methods [42, 44, 55, 58] usually estimate the rPPG signal from the original video frame and utilize attention mechanism [9, 23] to highlight high-quality facial regions. To

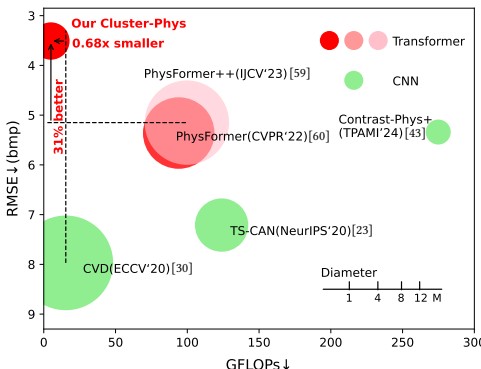

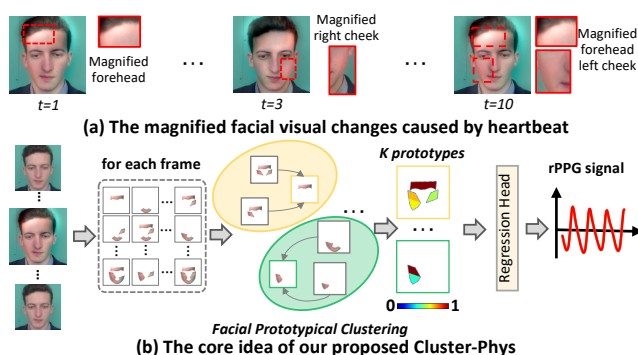

Figure 1: Performance and complexity evaluation for cross-dataset testing on MMSE-HR dataset . The proposed Cluster-Phys achieves 31% performance gain on RMSE than Phys-Former++ [59] while being 0.68× smaller than CVD [30]. The diameter of the circle indicates the model size.

Figure 2: (a) Illustration of magnified facial region changes caused by heartbeats in video-based rPPG signal estimation. (b) The core idea of the proposed Cluster-Phys. We aim to capture the representative prototypes containing semantically similar instances of subtle changes in each frame.

reduce the high computation costs of 3D-CNN, the spatial-temporal map is proposed, and the researchers [26, 27, 30, 31, 38] focus on the design of various architecture to embedding the spatial-temporal maps for better alignment with rPPG signals. To increase the robustness of rPPG estimation, some methods [26, 30] attempt to disentangle the physiological information with non-physiological features from the spatial-temporal map to obtain distilled physiological features. Recently, Transformer [46] is introduced for this field in terms of capturing the spatiotemporal contexts of rPPG [24, 59, 60]. In fact, because different physiological tissue structures, capillary density, and blood flow exist in different facial regions, their light absorption is not constant. This brings huge challenges to modeling the rPPG signals.

Considering the subtle changes in the face are imperceptible by the human eye, we magnified the video frame by the magnification methods [48, 51], thereby better revealing these subtle changes. As shown in Fig. 2 (a), we can see that different facial regions absorb light differently and exhibit different reactions over time. For example, in the first frame, the forehead exhibits a high reaction (*i.e.*, bright) while the right cheek exhibits a high reaction in the 3rd frame. The video-based rPPG estimation aims to capture these subtle changes from all frames. Intuitively, this motivates us to first model the light absorption of different facial regions in each frame and then capture the periodically changing rPPG signals by modeling across frames. To this end, as shown in Fig. 2 (b), we attempt to build compact and representative *facial prototypes* for each frame to represent the predominant light absorption from different facial regions. Then, the facial prototypes from all frames are used for rPPG estimation. Such a facial prototype-based method enables us to achieve superior performance while using much fewer parameters. As shown in Fig. 1, our method surpasses the previous SOTA method PhysFormer++ [59] by 31% in terms of the RMSE metric and the GFLOPs number of our method is ×68% smaller than CVD [30].

Specifically, we propose a novel rPPG-based physiological signal measurement framework named Cluster-Phys, which aims to capture the latent periodic physiological signals from all frames

through progressive facial ROI prototypical clustering (FPC). As shown in Fig. 3, in the facial ROI prototypical clustering module (§ 3.2), we aim to iteratively build compact and representative facial prototypes using a "*learn-and-cluster*" manner. In the *learn* phase, we first *learn* a soft assignment matrix through semantic similarity distance measurement between initialized prototypes and facial ROI features. In the *cluster* phase, we use the soft matrix to adaptively cluster representative facial ROI prototypes. Considering that facial physiological signals change over time, we devise a prototype updating strategy based on the semantic similarity between prototypes and ROI features to update the prototypes. The prototype updating strategy updates facial ROI prototypes of all frames, making it better capture global physiological signal changes for rPPG estimation. The established prototypes are processed in our spatial-temporal prototype interaction module (§ 3.3) where we focus on mining spatial and temporal correlations among prototypes to further enrich them. Considering that rPPG estimation necessitates modeling spatial changes in facial regions, and the temporal smoothness and periodic pattern consistency in rPPG signals, we separately use spatial and temporal interactions to perceive the spatial correlations among prototypes within each frame and periodic pattern consistencies. We recurrently perform the above prototype clustering and interaction processes to build the compact facial representation. Finally, we use the rPPG regression head (§ 3.4) to predict the rPPG signal based on the compact and representative prototypes learned above.

In summary, our contributions can be summarized as follows:

- To mine informative facial ROI clues for remote physiological measurement, we propose a Facial ROI Prototypical Clustering method to progressively learn and utilize representative ROI prototypes. To the best of our knowledge, it is the first attempt at prototypical clustering in this field.
- To capture the spatiotemporal contexts in ROI prototypes, we propose a spatial-temporal prototype interaction module that performs interaction on spatial and temporal separately.

- Extensive experiments show that our method achieves state-of-the-art performance on the four benchmark datasets referring to both intra-dataset and cross-dataset testing.

## 2 Related Work

**Deep learning-based rPPG Measurement.** In the early years, CNN-based methods [8, 41] with native backbone [37] are proposed to capture time-frequency representation for rPPG prediction. To capture the dynamic changes in temporal contexts, attention-based CNN method [2, 23, 44] and CNN-RNN based methods [15, 57] are proposed capture spatiotemporal features to eliminate the negative effect of head movements. To enhance the robustness of the rPPG predictor against unseen noises, Generative Adversarial Networks (GANs)-based methods [26, 38] are proposed to improve the distinguishability of the rPPG predictor. Recently, with the great success of transformer [17, 18, 52, 53, 63], Transformer-based methods [24, 60] have been proposed to aggregate long-range spatiotemporal features for rPPG estimation. Different from the above approaches, the proposed Cluster-Phys employs a facial ROI prototypical clustering method to build representative facial ROI prototypes, and uses a spatiotemporal interaction module to mine facial cues in the prototypes for rPPG estimation.

**Prototypical Clustering for Video Feature Aggregation.** Representative and compact feature representation is the pursuit of video understanding [7, 11, 12, 19, 62], such as group activity recognition [19], video retrieval [11]. Among these works, prototypical clustering aims to capture representative embeddings for groups of semantically similar instances. For example, Li *et al.* [19] designed a clustered attention transformer that captures both intra- and inter-group relations, thereby constructing better group informative features for group action recognition. In this paper, we focus on rPPG measurement from facial videos. Existing studies in this domain primarily concentrate on modeling subtle and sensitive physiological signals [2, 16, 58, 60], and mainly design models based on the original video feature sequences. We hypothesize that prototypical clustering can facilitate the extraction of crucial clues related to rPPG, and endeavor to verify this hypothesis in this field.

## 3 Methodology

### 3.1 Overview

Let an RGB facial video as $\mathbf{X}_v \in \mathbb{R}^{T \times H \times W \times 3}$, where $T$, $H$, and $W$ denote frame number, height, width of the video, respectively. The quasi-periodic pulse signal originates from subtle light reflections of blood vessels under the skin. The remote physiological measurement task aims to predict one-dimensional rPPG signal $s_{pre} \in \mathbb{R}^T$ that can reflect the quasi-periodic heartbeat from facial videos. Considering the motivation outlined in the introduction, *i.e.*, facial ROI is composed of different facial regions and each ROI has different physiological signals, we build the prototypes for the facial ROIs of each frame to learn representative embedding of semantically similar physiological information. To capture the spatial correlations between prototypes with each frame and the periodic variation in the temporal dimension, we design the spatiotemporal prototype interaction to capture its information separately. As illustrated in Fig. 3, we first transform the video $\mathbf{X}_v$ into an ROI-based MSTmap $\mathbf{X}_{map} \in \mathbb{R}^{T \times N \times 6}$, where $N$ denotes the detected facial ROI number.

Then, we embed the MSTmap $\mathbf{X}_{map}$ to high-dimensional feature $\mathbf{X} \in \mathbb{R}^{T \times N \times D}$. Subsequently, we take the $\mathbf{X}$ as the input and design a facial ROI prototypical clustering strategy to build prototypes. Then, facial ROI prototypes are fed into the spatial-temporal prototype interaction module to learn the global spatial and temporal correlation of rPPG clues. Finally, we use an rPPG regression head to predict the one-dimensional rPPG signal $s_{pre} \in \mathbb{R}^T$.

### 3.2 Facial ROI Prototypical Clustering

In this field, the gap between facial ROI-level features and latent rPPG clues has not been completely elaborately explored. Intuitively, physiological information implicit in facial ROIs is closely related to the rPPG signal. In this work, we focus on the modeling of facial ROI information and discovering crucial semantics from subtle differences in faces. The rPPG estimation is based on the periodic changes in optical absorption of a local tissue with changes in blood volume, corresponding to the heartbeats. Therefore, the facial ROI will exhibit periodic variation within the time cycles. In addition, the facial ROI is composed of different facial regions and each region has different physiological signals. To capture this periodic variation for accurate rPPG estimation, we build the prototypes for the facial ROIs of each frame to learn representative embedding of semantically similar instances.

*3.2.1 Facial ROI Prototype Initialization.* Our facial ROI prototype initialization includes two steps, *i.e.*, facial ROI feature preparation and prototype initialization.

**Facial ROI feature Preparation**. Compared with the background, the face contains wealthy rPPG-related physiological information. In addition, since blood flow in blood vessels under the skin varies in different facial areas, different facial areas need to be analyzed separately. In our work, we detect facial landmarks of the video and divide the whole facial region into multiple facial ROI as MSTmap[27, 29, 30]. Specifically, we detect 6 meta-ROI (*i.e.*, forehead, left upper cheek, left lower cheek, right upper cheek, right lower cheek, and chin) of the face, and generate $N = (2^6 - 1) = 63$ informative ROI combination blocks. Then, the average pixel values of each facial ROI are concatenated along the temporal dimension to build the MSTmap $\mathbf{X}_{map} \in \mathbb{R}^{T \times N \times 6}$, where 6 represents {R,G,B,Y,U,V} color channels. Finally, we embed the MSTmap $M$ to high-dimensional feature $\mathbf{X} \in \mathbb{R}^{T \times N \times D}$ using a fully connected layer as the ROI feature representation.

**Prototype Initialization.** In view of the previous discussion of motivation, here we implement facial clue clustering. Given the input ROI feature embeddings $\mathbf{X} \in \mathbb{R}^{T \times N \times D}$, we adopt the density peaks clustering algorithm to initialize facial ROI prototypes as $\mathbf{C} \in \mathbb{R}^{T \times K \times D}$, where $K = N \times \rho$ is the number of facial ROI prototypes, and $\rho$ is the cluster sparse ratio hyper-parameter. Specifically, for the input feature embeddings $\{x_i\}_{i=1}^N \in \mathbb{R}^{N \times D}$ of each frame, we first calculate its local density by:

$$\varphi_i = \exp(-\frac{1}{k} \sum_{x_j \in \text{KNN}(x_i)} \sigma(x_i, x_j)), \qquad (1)$$

where $\sigma(\cdot, \cdot)$ denotes the Euclidean distance, KNN $(x_i)$ denotes the $k$-nearest neighbors of $x_i$. Subsequently, we compute the relative

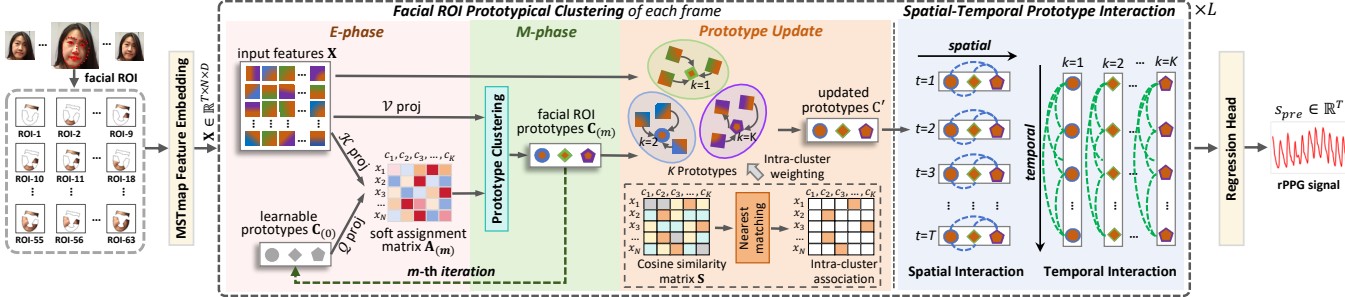

**Figure 3: Overview of our Cluster-Phys. § 3.2 Facial ROI Prototypical Clustering (FPC). We first build the Facial ROI cluster using a "learn-and-cluster" manner to iteratively _learn_ semantic correlations between prototypes and features and _cluster_ prototypes. § 3.3 Spatio-Temporal Prototype Interaction. We separately mine spatial correlations and temporal periodic clues of rPPG signals to enrich prototypes. § 3.4 rPPG Signal Estimation. The learned prototypes are used for rPPG estimation.**

distance indicator between $\varphi_i$ and $\varphi_j$ as follows:

$$\delta_i = \begin{cases} \min_{j:\varphi_j > \varphi_i} \sigma(x_i, x_j), & \text{if } \exists j \text{ s.t. } \varphi_j > \varphi_i, \\ \max_j \sigma(x_i, x_j), & \text{otherwise.} \end{cases} \tag{2}$$

Intuitively, $\varphi_i$ represents the local density around $x_i$, and $\delta_i$ represents $x_i$'s distance to other high-density features. In order to select the appropriate initial prototype (or cluster center), we use the product method to simultaneously consider the local density $\varphi_i$ of features $x_i$ and its degree $\delta_i$ of isolation from other dense regions. We define the $x_i$'s score as $\varphi_i \times \delta_i$, and select representative top-$K$ of them as the learnable facial ROI prototypes $\mathbf{C}_{(0)} \in \mathbb{R}^{T \times K \times D}$.

*3.2.2 Prototype Clustering.* After getting the learnable prototypes $\mathbf{C}_{(0)}$ in prototype initialization, we perform prototype clustering to aggregate rPPG-related information from facial ROI features and build a prototype center. Considering that we aim to mine rPPG-related information from the input feature $\mathbf{X} \in \mathbb{R}^{T \times N \times D}$, we use the Expectation-Maximization (E-M) approach cluster prototype in $M$ iterations. As shown in Fig. 3, in $E$ phase, we first construct a soft assignment matrix $\mathbf{A}_{(m)}$ to model the correlation between the initial prototypes $\mathbf{C}_{(0)} \in \mathbb{R}^{T \times K \times D}$ and the input feature $\mathbf{X} \in \mathbb{R}^{T \times N \times D}$:

$$E\text{-phase: } \mathbf{A}_{(m)} = \text{softmax}(Q^{C_{(m)}}(\mathcal{K}^X)^\top)_{\mathcal{K}}, \tag{3}$$

where $m \in \{1, \cdots, M\}$, $Q^C \in \mathbb{R}^{T \times K \times D}$ denotes the query vector projected from the facial ROI prototype $\mathbf{C}$, and $\mathcal{K}^X \in \mathbb{R}^{T \times N \times D}$ correspond to the key vectors projected from the input features $\mathbf{X}$. The soft assignment matrix globally represents the correlation between prototype and feature, unlike DPC-KNN which only updates clustering using hard-matching with the features of prototype.

Based on the soft assignment matrix $\mathbf{A}_{(m)}$ from the $E$ phase, we use it to cluster facial ROI prototypes in $M$ phase:

$$M\text{-phase: } \mathbf{C}_{(m+1)} = \mathbf{A}_{(m)} \mathcal{V}^X \in \mathbb{R}^{T \times K \times D}, \tag{4}$$

where $\mathcal{V}^X \in \mathbb{R}^{T \times N \times D}$ indicates the value vectors projected from the input features $\mathbf{X}$. In each iteration $m$, this prototype clustering strategy iteratively _learns_ the soft assignment matrix $\mathbf{A}_{(m)}$ in $E$ phase and use it to _cluster_ the facial ROI prototypes $\mathbf{C}$ in $M$ phase. This dynamic "_learn-and-cluster_" strategy can continuously learn the soft assignment matrix between prototypes and the facial ROI features to build better prototypes.

*3.2.3 Prototype Update.* Based on the aforementioned prototype clustering strategy, we obtained the facial ROI prototypes by considering the soft assignment correlation between input features and initial learnable prototypes. However, considering that facial physiological signals change over time [34, 39], there will be significant differences between facial ROI prototypes at different times. Therefore, we need to make the prototype more representative of facial ROI prototypes at all times so that it can better capture global physiological signal changes for rPPG estimation. To this end, we propose the prototype updating strategy based on the semantic similarity between prototypes and input features. Specifically, for each time $t$, we use the cosine similarity to calculate the semantic similarity $\mathbf{S} \in \mathbb{R}^{N \times K}$ between the facial ROI prototypes $\mathbf{C}_t \in \mathbb{R}^{K \times D}$ and input features $\mathbf{X}_t \in \mathbb{R}^{N \times D}$:

$$\mathbf{S} = \frac{\mathbf{X}_t \cdot \mathbf{C}_t^\top}{\|\mathbf{X}_t\| \|\mathbf{C}_t\|}, \tag{5}$$

where $\mathbf{S}_{i,j}$ denotes the similarity between $i$-th features $x_i$ in $\mathbf{X}_t$ and $j$-th prototype $c_j$ in $\mathbf{C}_t$. To expand the influence of semantic closer input features to the prototypes while also avoiding potential flaws from irrelevant features, we apply the nearest matching strategy to divide the input features into different prototype clusters and update the prototypes based on the intra-cluster associations. We define $Set(c_j)$ as the intra-cluster features, which denote the set of input features assigned to the prototype $c_j$. Next, we use a similarity-based weighting fusing method to update the prototype from the clustering level. For intra-cluster features, features close to the prototype should have a greater impact on clustering and require greater weight. Therefore, we calculate the weights of prototype and intra-cluster features respectively according to the semantic similarity $\mathbf{S}$:

$$\begin{aligned} W_i &= \frac{\exp(\mathbf{S}_{i,j})}{\sum_{x_i \in Set(c_j)} \exp(\mathbf{S}_{i,j}) + e}, \\ W_j &= \frac{e}{\sum_{x_i \in Set(c_j)} \exp(\mathbf{S}_{i,j}) + e}. \end{aligned} \tag{6}$$

For each $c_j$ in $\mathbf{C}_t$, we use weights $W_i$ and $W_j$ to update its feature:

$$c_j' = W_j \cdot c_j + \sum_{i=1}^N W_i \cdot x_i. \tag{7}$$

Finally, for each frame $t$, we can get the updated facial ROI prototype $\mathbf{C}_t' = \{c_i'\}_{i=1}^K \in \mathbb{R}^{K \times D}$. Therefore, the video-level prototype

representation is $\mathbf{C}' \in \mathbb{R}^{T \times K \times D}$. To gradually mine informative rPPG clues to update the prototypes, we perform $L$ steps for facial ROI prototypical hierarchically. The spatial-temporal interaction module is followed by each facial ROI prototypical clustering block.

## 3.3 Spatial-Temporal Prototype Interaction

After the facial ROI prototypical clustering mentioned above, we obtained the compact prototypes $\mathbf{C}'$ that enhanced the representation of the facial ROI. Considering that rPPG estimation necessitates spatial-temporal modeling changes in facial signals, it is required to learn the spatial correlation within these facial ROI prototypes. Furthermore, to maintain temporal smoothness and periodic pattern consistency in video-based rPPG estimation, temporal interactions of facial ROI prototypes are also required. We employ a Prototype Interactor to capture both spatial and temporal contextual information. 1). For *spatial-wise interaction*, we first divide $\mathbf{C}'$ into $\mathbf{C}'_{spatial} = \{\mathbf{c}'^{(t)} \in \mathbb{R}^{K \times D} | t = 1, \ldots, T\}$, then feed it into the **ProtoInter(.)** operator to model the spatial correlation between $K$ prototypes at each frame. 2). Similarly, for *temporal-wise interaction*, we convert $\mathbf{C}'$ into $\mathbf{C}'_{temporal} = \{\mathbf{c}'^{(k)} \in \mathbb{R}^{T \times D} | k = 1, \ldots, K\}$, then feed it into the **ProtoInter(.)** operator to model temporal correlation between $T$ frames of each prototype. Suppose the facial ROI prototype features $\mathbf{C}' \in \mathbb{R}^{T \times K \times D}$. The **ProtoInter** can be formulated as follows. The input $\mathbf{C}'$ ($\mathbf{C}'_{spatial}$ and $\mathbf{C}'_{temporal}$) of Eq. 8 is determined by spatial-wise or temporal-wise operation.

$$\mathbf{C}'' = \text{ProtoInter}(\mathbf{C}') \Leftrightarrow$$
$$\begin{cases} \hat{\mathbf{C}} = \text{MSA}(\text{LN}(\mathbf{C}')) + \mathbf{C}'; \\ \mathbf{C}'' = \text{FFN}(\text{LN}(\hat{\mathbf{C}})) + \hat{\mathbf{C}}, \end{cases} \quad (8)$$

where MSA, LN, and FFN are multi-head attention, layer norm and feed-forward layer in [46].

## 3.4 Model Training

From the entire Facial ROI Prototype modeling, we obtain the final facial ROI prototypes $\mathbf{C}'' \in \mathbb{R}^{K^{(L)} \times T \times D}$. Next, we use an rPPG regression head consisting of a spatial average pooling layer and a linear projection layer to predict 1D rPPG signal $s_{pre} \in \mathbb{R}^T$ based on the prototype features $\mathbf{C}''$. Following the practice [26, 30, 60], the standard Negative Pearson Correlation loss is used to minimize error between $s_{pre}$ and ground truth rPPG signals $s_{gt} \in \mathbb{R}^T$:

$$\mathcal{L}_{rPPG} = 1 - \frac{Cov(s_{pre}, s_{gt})}{\sqrt{Cov(s_{pre}, s_{pre})}\sqrt{Cov(s_{gt}, s_{gt})}}, \quad (9)$$

where $Cov(x, y)$ denotes the covariance of variables $x$ and $y$.

## 4 Experiments

### 4.1 Experimental Setup

**Datasets. UBFC-rPPG** [1] is a commonly used pure dataset for physiological estimation. It contains 42 facial videos of 42 participants. **PURE** [1] contains 60 facial videos from 10 subjects. Each person records 6 videos in 6 different scenarios. **VIPL-HR** [29] is a challenging large-scale dataset for rPPG estimation. It records 2,378 facial videos from 107 subjects under 9 complicated and diverse

scenarios, such as different head motions and illumination conditions. **MMSE-HR** [45] has 102 videos captured from 40 subjects of different races with diverse facial expressions.

**Implementation Details.** For HR estimation, following previous work [24, 60], we apply a 1st-order Butterworth filter to convert the rPPG signal into an HR value with a cutoff frequency range of [0.75Hz, 2.5Hz], corresponding to [45, 150] beats per minute. Subsequently, we perform the PSD [50] to estimate HR values for video segments. Finally, we get a video-level HR by averaging HR values from all segments.

### 4.2 Intra-dataset Testing

**HR Estimation on UBFC-rPPG.** We first evaluate the proposed method on the UBFC-rPPG dataset, which has simple scenarios and high-quality data. As illustrated in Table 1, the proposed method outperforms both traditional and deep learning-based methods. Compared with state-of-the-art Dual-GAN [26], our method significantly reduces MAE and RMSE by 43% and 25%, respectively. Furthermore, compared to the recent supervised version of Contrast-Phys+ [43] based on contrastive loss, we have achieved a leading advantage in most metrics. These noteworthy advancements underscore the effectiveness of our clustering strategy in capturing informative rPPG clues, leading to a notable improvement in HR estimation accuracy.

**HR Estimation on PURE.** Unlike the UBFC-rPPG dataset, the extensive head movements in PURE significantly disrupt rPPG acquisition, posing significant challenges. Nevertheless, our model achieves substantial improvements, such as a 53% lower RMSE compared to Dual-GAN [26] and a 62% reduction compared to Li *et al.* [22]. These results indicate that our clustering strategy effectively consolidates crucial facial signals even in complex environments, successfully minimizing the impact of external disturbances.

**HR Estimation on VIPL-HR.** The large-scale VIPL-HR dataset is known for its extremely complex external noise challenges, such as head movements, talking, and dark and bright scenarios. Following the protocol in [27, 29, 60], we conduct the subject-exclusive 5-fold cross-validation. As shown in Table 1, we can see that our model surpasses previous state-of-the-art NEST (MAE of 4.76 bpm and RMSE of 7.51 bpm) by a large margin, indicating the effectiveness of Cluster-Phys in mitigating the impact of various noises. While noise-induced variation may be more pronounced than rPPG signals, our aggregation strategy allows the model to focus on more high-quality facial cues and filter the noisy information.

**HRV Estimation on UBFC-rPPG.** In addition to HR estimation, we also conduct the heart rate variability (HRV) and respiration frequency (RF) estimation. HRV and RF estimations require accurately measured high-quality rPPG signals. As shown in Table 2, we can see that the proposed approach outperforms the existing state-of-the-art traditional methods by a large margin.

### 4.3 Cross-dataset Testing

Cross-dataset testing is essential for assessing the model's generalization capabilities in unseen scenarios. Following previous works [24, 26, 60], we conduct three cross-dataset tests . The experimental results are reported in Table 3.

| | Method | Venue | UBFC-rPPG | | | PURE | | | VIPL-HR | | |
|---|---|---|---|---|---|---|---|---|---|---|---|
| | | | MAE↓ | RMSE↓ | r ↑ | MAE↓ | RMSE↓ | r ↑ | MAE↓ | RMSE↓ | r ↑ |
| *Traditional* | GREEN [47] | Optics Express'08 | 7.50 | 14.41 | 0.62 | 7.23 | 17.05 | 0.69 | - | - | - |
| | ICA [34] | Optics Express'10 | 5.17 | 11.76 | 0.65 | 3.76 | 12.60 | 0.85 | - | - | - |
| | CHROM [3] | TBE'13 | 2.37 | 4.91 | 0.89 | 2.07 | 9.92 | 0.99 | 11.4 | 16.9 | 0.27 |
| | 2SR [4] | PM'14 | 6.90 | 18.50 | 0.65 | 2.44 | 3.06 | 0.98 | - | - | - |
| | SAMC [45] | CVPR'16 | - | - | - | - | - | - | 15.9 | 21.0 | 0.24 |
| | POS [49] | TBE'16 | 4.05 | 8.75 | 0.78 | 0.80 | 4.11 | 0.98 | 11.5 | 17.2 | 0.24 |
| *Deep Learning-based* | SynRhythm [28] | ICPR'18 | 5.59 | 6.82 | 0.72 | - | - | - | - | - | - |
| | DeepPhys [2] | ECCV'18 | 2.90 | 3.63 | - | 0.83 | 1.54 | 0.99 | 11.0 | 13.8 | 0.72 |
| | PhysNet [57] | BMVC'19 | 2.95 | 3.67 | - | 1.90 | 3.44 | 0.98 | 10.8 | 14.8 | 0.20 |
| | RhythmNet [29] | TIP'19 | - | - | - | - | - | - | 5.30 | 8.14 | 0.76 |
| | CVD [30] | ECCV'20 | - | - | - | - | - | - | 5.02 | 7.97 | 0.79 |
| | Siamese-rPPG [44] | SAC'20 | 0.48 | 0.97 | - | 0.51 | 1.56 | 0.83 | - | - | - |
| | PulseGAN [38] | JBHI'21 | 1.19 | 2.10 | 0.98 | - | - | - | - | - | - |
| | Gideon *et al.* [6] | ICCV'21 | 1.85 | 4.28 | 0.93 | 2.30 | 2.90 | 0.99 | 9.01 | 14.02 | 0.58 |
| | Dual-GAN [26] | CVPR'21 | 0.44 | 0.67 | 0.99 | 0.82 | 1.31 | 0.99 | 4.93 | 7.68 | 0.81 |
| | PhysFormer [60] | CVPR'22 | - | - | - | - | - | - | 4.97 | 7.79 | 0.78 |
| | Contrast-Phys [42] | ECCV'22 | 0.64 | 1.00 | 0.99 | 1.00 | 1.40 | 0.99 | 32.1 | 36.1 | 0.04 |
| | TFA-PFE [16] | AAAI'23 | 0.76 | 1.62 | - | 1.44 | 2.50 | - | - | - | - |
| | SiNC [40] | CVPR'23 | 0.59 | 1.83 | 0.99 | 0.61 | 1.84 | **1.00** | - | - | - |
| | NEST [27] | CVPR'23 | - | - | - | - | - | - | 4.76 | 7.51 | **0.84** |
| | Li *et al.* [22] | ICCV'23 | 0.48 | 0.64 | **1.00** | 0.64 | 1.16 | 0.99 | 4.97 | 7.79 | 0.78 |
| | PhysFormer++ [59] | IJCV'23 | - | - | - | - | - | - | 4.88 | 7.62 | 0.80 |
| | Yue *et al.* [61] | TPAMI'23 | 0.58 | 0.94 | 0.99 | 1.23 | 2.01 | 0.99 | - | - | - |
| | Contrast-Phys+[43] | TPAMI'24 | **0.21** | 0.80 | 0.99 | 0.48 | 0.98 | 0.99 | - | - | - |
| | **Cluster-Phys(Ours)** | - | 0.25 | **0.50** | **1.00** | 0.37 | **0.61** | **1.00** | **4.07** | **6.79** | **0.84** |

**Table 1: Intra-dataset HR estimation results on the UBFC-rPPG, PURE, and VIPL-HR datasets. The best results are highlighted in bold, and the second-best results are underlined.**

| | Method | Venue | LF (n.u.) | | | HF (n.u) | | | LF/HF | | | RF (Hz) | | |
|---|---|---|---|---|---|---|---|---|---|---|---|---|---|---|
| | | | SD↓ | RMSE↓ | r ↑ | SD↓ | RMSE↓ | r ↑ | SD↓ | RMSE↓ | r ↑ | SD↓ | RMSE↓ | r ↑ |
| *Trad.* | GREEN [47] | Optics Express'08 | 0.186 | 0.186 | 0.280 | 0.186 | 0.186 | 0.280 | 0.361 | 0.365 | 0.492 | 0.087 | 0.086 | 0.111 |
| | ICA [34] | Optics Express'10 | 0.243 | 0.240 | 0.159 | 0.243 | 0.240 | 0.159 | 0.655 | 0.645 | 0.226 | 0.086 | 0.089 | 0.102 |
| | POS [49] | TBE'16 | 0.171 | 0.169 | 0.479 | 0.171 | 0.169 | 0.479 | 0.405 | 0.399 | 0.518 | 0.109 | 0.107 | 0.087 |
| *DL-based* | CVD [30] | ECCV'20 | 0.053 | 0.056 | 0.740 | 0.053 | 0.065 | 0.740 | 0.169 | 0.168 | 0.812 | 0.017 | 0.018 | 0.252 |
| | Dual-GAN [26] | CVPR'21 | 0.034 | 0.035 | 0.891 | 0.034 | 0.034 | 0.891 | 0.131 | 0.136 | 0.881 | 0.010 | 0.010 | 0.395 |
| | Gideon *et al.* [6] | ICCV'21 | 0.091 | 0.139 | 0.694 | 0.091 | 0.139 | 0.694 | 0.525 | 0.691 | 0.684 | 0.061 | 0.098 | 0.103 |
| | Contras-Phys [42] | ECCV'22 | 0.050 | 0.098 | 0.798 | 0.050 | 0.098 | 0.798 | 0.205 | 0.395 | 0.782 | 0.055 | 0.083 | 0.347 |
| | Contrast-Phys+ [43] | TPAMI'24 | 0.025 | 0.025 | 0.947 | 0.025 | 0.025 | 0.947 | 0.064 | 0.066 | 0.963 | 0.029 | 0.029 | 0.803 |
| | **Cluster-Phys (Ours)** | - | **0.021** | **0.021** | **0.960** | **0.021** | **0.021** | **0.960** | **0.067** | **0.062** | **0.970** | **0.007** | **0.007** | **0.816** |

**Table 2: Heart Rate Variability (HRV) and Respiration Frequency (RF) estimation on the UBFC-rPPG dataset. LF, HF, and RF represent low frequency, high frequency, and respiration frequency, respectively. "n.u." denotes normalized units.**

**PURE→UBFC-rPPG.** Since the PURE dataset is much more complex than the UBFC-rPPG dataset, this adaption process could be relatively easy. Compared with state-of-the-art Dual-GAN [26], our method achieves further breakthroughs in all metrics. Notably, the RMSE of our method is lower than 1 bpm, indicating that the predicted HRs are closely aligned with the ground truth HRs.

**UBFC-rPPG→PURE.** From Table 3, we observe that all deep learning-based methods performed poorly, with RMSE exceeding 10 bpm. When compared to the complex to simple test (PURE→UBFC-rPPG), the significant performance degradation in this mode may be attributed to the introduction of head movement scenarios in the PURE dataset. In such situations, the methods struggle to extract rPPG signals from chaotic facial features, resulting in inaccurate HR estimations. In contrast, our approach attains an RMSE lower than 10 bpm and a Pearson *r* close to 1, significantly outperforming previous methods. These results show that our clustering strategy prefers to learn the intrinsic rPPG clues of face regions.

**VIPL-HR→MMSE-HR.** In this cross-dataset test, VIPL-HR contains different motion patterns and light changes, while MMSE-HR contains diverse facial expressions. As shown in Table 3, our method achieves the lowest MAE (1.74), lowest RMSE (3.51), and highest *r* (0.96), respectively. Compared to the EfficientPhys-T1 [24] and PhysFormer family [59, 60] based on spatial-temporal Transformer, our Cluster-Phys has significant advantages across all metrics. We attributed it to the proposed clustering strategy that can extract representative rPPG-aware prototypes, which shows robustness in this challenging cross-dataset test.

## 4.4 Ablation Studies

**Impact of Main Components.** As shown in Table 4, we investigate the main components including Facial ROI Prototypical Clustering (§ 3.2) and Spatial-Temporal Prototype Interaction (§ 3.3). From row 2, we can see that the model without prototype clustering leads to large performance degradation, *e.g.*, MAE is dropped from 3.47 to

| | Method | Venue | PURE → UBFC-rPPG | | | UBFC-rPPG → PURE | | | VIPL-HR → MMSE-HR | | |
|---|---|---|---|---|---|---|---|---|---|---|---|
| | | | MAE↓ | RMSE↓ | r↑ | MAE↓ | RMSE↓ | r↑ | MAE↓ | RMSE↓ | r↑ |
| *Traditional* | CHROM [3] | TBE'13 | 3.10 | 6.84 | 0.93 | 5.77 | 14.93 | 0.81 | - | 13.97 | 0.55 |
| | Li2014 [21] | CVPR'14 | - | - | - | - | - | - | - | 19.95 | 0.38 |
| | POS [49] | TBE'16 | 3.52 | 8.38 | 0.90 | 3.67 | 11.82 | 0.88 | - | - | - |
| | SAMC [45] | CVPR'16 | - | - | - | - | - | - | - | 11.37 | 0.71 |
| *Deep Learning-based* | DeepPhys [2] | ECCV'18 | 1.21 | 2.90 | 0.99 | 5.54 | 18.51 | 0.66 | - | - | - |
| | PhysNet [57] | BMVC'19 | 1.63 | 3.79 | 0.98 | 9.36 | 20.63 | 0.62 | - | 13.25 | 0.44 |
| | RhythmNet [29] | TIP'19 | - | - | - | - | - | - | - | 7.33 | 0.78 |
| | CVD [30] | ECCV'20 | - | - | - | - | - | - | 5.02 | 7.97 | 0.79 |
| | TS-CAN [23] | NeurIPS'20 | 1.30 | 2.87 | 0.99 | 3.69 | 13.80 | 0.82 | 3.85 | 7.21 | 0.86 |
| | Dual-GAN [26] | CVPR'21 | 0.74 | 1.02 | 0.99 | - | - | - | - | - | - |
| | Contrast-Phys [42] | ECCV'22 | 10.22 | - | 0.45 | 19.61 | - | 0.33 | - | - | - |
| | PhysFormer [60] | CVPR'22 | - | - | - | - | - | - | 2.84 | 5.36 | 0.92 |
| | EfficientPhys-C [24] | WACV'23 | 2.13 | 3.00 | 0.99 | 5.47 | 17.04 | 0.71 | 2.91 | 5.43 | 0.92 |
| | EfficientPhys-T1 [24] | WACV'23 | 3.83 | 5.62 | 0.87 | - | - | - | 3.48 | 7.21 | 0.86 |
| | SiNC [40] | CVPR'23 | 6.64 | - | 0.59 | 4.02 | - | 0.86 | - | - | - |
| | Li *et al.* [22] | ICCV'23 | 0.71 | 1.45 | 0.99 | - | - | - | - | - | - |
| | PhysFormer++ [59] | IJCV'23 | - | - | - | - | - | - | 2.71 | 5.15 | 0.93 |
| | **Cluster-Phys (Ours)** | - | **0.61** | **0.95** | **1.00** | **3.08** | **7.35** | **0.96** | **1.74** | **3.51** | **0.96** |

**Table 3: Cross-dataset testing results on the PURE →UBFC-rPPG, UBFC-rPPG→PURE, and VIPL-HR →MMSE-HR.**

| | Proto. Clustering | | Proto. Interaction | | MAE↓ | RMSE↓ | r↑ |
|---|---|---|---|---|---|---|---|
| | Cluster | Update | Spatial | Temporal | | | |
| 1 | – | – | ✓ | ✓ | 3.87 | 6.95 | 0.79 |
| 2 | ✓ | – | ✓ | ✓ | 3.55 | 6.41 | 0.84 |
| 3 | ✓ | ✓ | – | – | 4.06 | 7.35 | 0.78 |
| 4 | ✓ | ✓ | ✓ | – | 3.70 | 6.73 | 0.83 |
| 5 | ✓ | ✓ | – | ✓ | 3.61 | 6.68 | 0.83 |
| 6 | ✓ | ✓ | ✓ | ✓ | **3.47** | **6.23** | **0.85** |

**Table 4: Ablation studies of the main components on the VIPL-HR dataset.**

| Clustering Strategy | MAE↓ | RMSE↓ | r↑ |
|---|---|---|---|
| Random | 4.04 | 7.30 | 0.80 |
| K-means | 3.88 | 6.92 | 0.81 |
| DPC [5] | 3.65 | 6.57 | 0.84 |
| **FPC (Ours)** | **3.47** | **6.23** | **0.85** |

**Table 5: Ablation studies of clustering strategy on the VIPL-HR dataset. FPC denotes the proposed Facial ROI Prototypical Clustering.**

3.87. This result reflects that facial prototype clustering can capture crucial rPPG-related clues for rPPG estimation. When incorporating the prototype cluster, the model will perform better as it learns better features from the cluster. From rows 3-5, we can conclude that both the spatial prototype interaction and temporal prototype interaction can help the model learn the spatial and temporal correlations for boosting rPPG estimation. Furthermore, we employ Bland-Altman plots to analyze the data consistency between our predicted and ground-truth HR. As shown in Fig. 6, we can see that our facial ROI prototypical clustering strategy contributes to better data consistency, *i.e.*, smaller standard deviation. Since the evaluated VIPL-HR dataset has extreme noise, such as head movements and light flickering, therefore the model without FPC leads to a significant standard deviation from the ground truth.

**Impact of Clustering Strategy.** We evaluate different clustering strategies, *i.e.*, "Random", "K-means", "DPC", and the proposed "FPC". As shown in Table 5, we can find that the "Random" cluster strategy performed worst because it randomly clusters all ROIs, making it hard for the model to learn the rPPG signals. The K-means strategy performs slightly better than Random, as it clusters ROIs based on feature similarity. In contrast, the DPC strategy is inferior to our method, as it predominantly relies on local density and does not perform well with facial ROI features characterized by high density. It's evident that the proposed FPC performs the best, we attribute to the FPC can iteratively cluster effective facial ROI prototypes guided by semantic similarity.

## 4.5 Qualitative Results

**(1) Visualization of Feature Clustering.** As depicted in Fig. 4, we visualize the prototypes in our facial ROI prototypical clustering model. Given a video, in the 1st stage, 63 meta-ROIs are clustered into 32 prototypes, we can see that (a1) captures the forehead and chin, while (a2) captures the left cheek, the right cheek, and the chin. In the 2nd stage, (b) captures the most representative ROIs, *i.e.*, the left cheek and the right cheek. Similarly, in the 3rd and 4th stages, (c) and (d) capture the left and right cheek. The facial prototypical clustering strategy adaptively aggregates the most representative ROIs to construct compact prototypes for rPPG estimation. **(2) Robust HR Estimation Cases.** We visualize four complex scenarios in Fig. 5 to verify the robustness of the model. From examples (a)-(c), we can obviously observe that not only on rPPG periodicity but also on the continuity and smoothness of the rPPG curve, our Cluster-Phys performs much better than PhysFormer [60]. It indicates the good robustness and accuracy of our method. In addition, we also give an extremely complex rPPG measurement scenario with both head movement and occlusion noises in Fig. 5 (d). Despite the challenges we encountered with our approach, it still outperforms PhysFormer.

## 4.6 Model Complexity Analysis

As shown in Table 6, we give detailed statistics on the complexity of the model of our method and open-source methods. Compared to the CNN-based method EfficientPhys-C, our method significantly outperforms it in RMSE with lower GFLOPs and fast inference

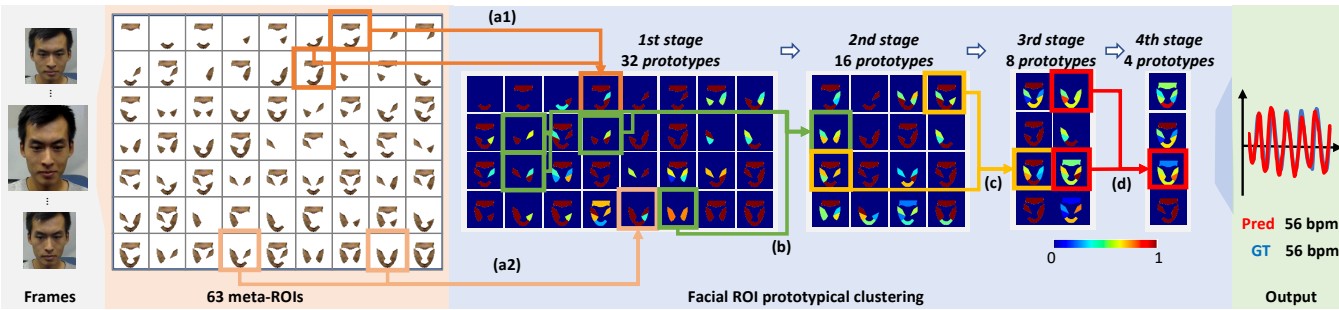

**Figure 4: Visualization of cluster-based feature aggregation on the VIPL-HR dataset. In each stage, the facial ROI prototype clustering (FPC) module adaptively aggregates the most representative ROIs for HR estimation.**

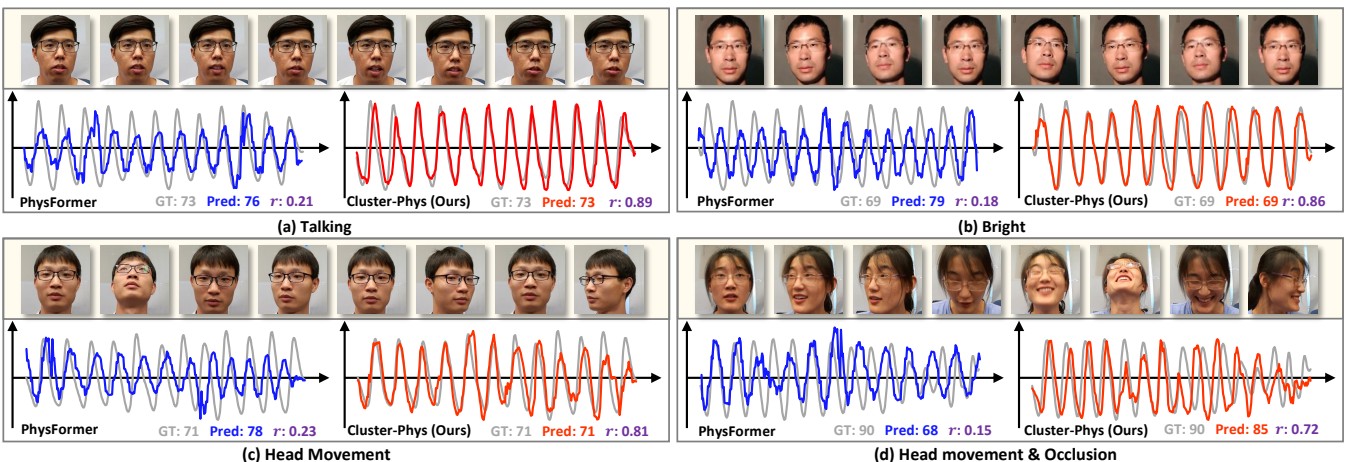

**Figure 5: The comparison of prediction results with ground truth under different complex scenarios on the VIPL-HR dataset. Compared with Physformer, our method predicts smoother and more accurate rPPG signals.**

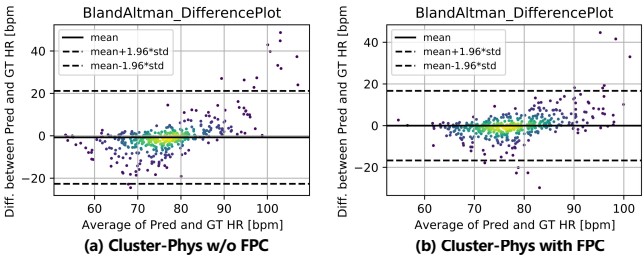

**Figure 6: The BlandAltman plots show the data consistency between predicted and ground-truth HR on the VIPL-HR dataset.**

| Method | Arch. | GFLOPs↓ | #Param.↓ | Infer.↓ | RMSE↓ |
|---|---|---|---|---|---|
| PhysNet [57] | CNN | 130.38 | **0.73** | **33** | 13.25 |
| TS-CAN [23] | CNN | 123.92 | 3.91 | 98 | 7.21 |
| CVD [30] | CNN | 15.34 | 12.34 | 86 | 7.97 |
| EfficientPhys-C [24] | CNN | 62.64 | 3.84 | 71 | 7.21 |
| Contrast-Phys+ [43] | CNN | 274.85 | 0.86 | 1215 | 5.34 |
| EfficientPhys-T1 [24] | Trans. | 106.09 | 6.87 | 5486 | 5.91 |
| PhysFormer [60] | Trans. | 94.02 | 7.03 | 109 | 5.36 |
| PhysFormer++ [59] | Trans. | 99.70 | 9.79 | - | 5.15 |
| **Cluster-Phys (Ours)** | Trans. | **4.97** | 1.92 | 82 | **3.51** |

**Table 6: The comparisons of Model GFLOPs, Parameters (M), and Inference Latency (ms).**

speed. Similarly, compared to the Transformer-based method Phys-Former, our method consumes only 5.28% GFLOPs and achieves better performance. In summary, these results validate the lower complexity and superior performance of our method.

## 5 Conclusion

In this paper, we propose Cluster-Phys, a novel facial prototypical aggregation architecture for remote physiological measurement. We propose a Facial ROI Prototypical Clustering (FPC) module to progressively mine and aggregate facial ROI clues to build prototypes. To capture the spatial correlations among prototypes within each frame and ensure the temporal periodic pattern consistency in rPPG signals, we devise a spatial-temporal prototype interaction module that separately perceives these correlations and consistencies. Extensive experiments conducted on both intra-dataset and cross-dataset tests show that our Cluster-Phys achieves new state-of-the-art performance.

# Acknowledgments

This work was supported the National Natural Science Foundation of China (62272144,72188101,62020106007 and U20A20183), the Major Project of Anhui Province (202203a05020011), and the Fundamental Research Funds for the Central Universities (JZ2024HGTG0309, JZ2024AHST0337, JZ2023YQTD0072, and 226-2022-00051).

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
