# OpenReview forum: "Cluster-Phys: Facial Clues Clustering Towards Efficient Remote Physiological Measurement"
_acmmm.org/ACMMM/2024/Conference — MM2024 Oral_

### Official Review · Reviewer_PLxG · 2024-05-20

**Rating:** 4
**Confidence:** 4

**Summary:**

The paper proposes a novel rPPG-based physiological signal measurement framework named Cluster-Phys, which aims to capture latent periodic physiological signals across all frames through progressive facial ROI prototype clustering. The effectiveness of the proposed method is validated through experiments in Intra-dataset Testing and Cross-dataset Testing.

**Strengths:**

1. The experiments are thorough and show excellent results, achieving better performance with less computational cost.
2. Code release is promised.
3. The proposed method based on prototypical clustering can inspire other researchers in the community.

**Limitations:**

1. The selection of facial ROIs is crucial. It is currently unclear how 63 ROI combination blocks are derived from the 6 meta-ROIs. Additionally, why are these 6 meta-ROI regions selected as the criteria? How would other ROI criteria, such as facial action unit ROIs, perform? These ROI selections might provide a more detailed facial region analysis.
2. The description is unclear. In lines L268 and L322 of the manuscript, it is not clear what D represents in T × N × D.
3. The manuscript lacks a deeper discussion of the method, such as its 'limitations'.

**Suitability:**

2

---

### Official Review · Reviewer_zZnQ · 2024-06-04

**Rating:** 3
**Confidence:** 2

**Summary:**

This paper proposes a novel clustering-driven remote physiological measurement framework (Cluster-Phys), which employs a facial ROI prototypical clustering module to adaptively cluster the representative facial ROI features as facial prototypes and then update facial prototypes with highly semantic correlated base ROI features. Experimental results on four public datasets illustrate that the proposed method is superior to these existing methods.

**Strengths:**

Compared with most existing methods, this work proposes a new rPPG measurement framework, Cluster-Phys, which is a facial ROI Prototypical Clustering method to progressively learn and utilize representative ROI prototypes. The authors also mentioned that it was the first attempt at prototypical clustering in this field. This idea sounds novel. And extensive experiments have been conducted to validate the performance of the proposed method.

**Limitations:**

1. Although the framework of the proposed method (Cluster-Phys) is drawn clearly and a lot of experiments are implemented to illustrate the performance of Cluster-Phys, the descriptions for Cluster-Phys are not clear. The writing should be earnestly modified. Additionally, in the first page, Fig.1 is given but the introduction for Fig.1 is missed in the main context.
2. In Section 3.3, how to achieve the prototype interaction? Give more details for this.
3. In Section 3.2, is the feature of each input image updated with the network learning? How to set the number of prototype clustering?
3. In Fig.6, why are the sizes of inputted figures consistent? The size of the middle figure is bigger than others.

**Suitability:**

3

---

### Official Review · Reviewer_iS6m · 2024-06-04

**Rating:** 4
**Confidence:** 3

**Summary:**

This presents a framework for remote physiological measurement using video-based analysis of facial regions. This work proposes to use prototypical clustering, focusing on facial regions of interest (ROIs) to capture the subtle changes in the face. Experiments are done on both intra-dataset and cross-dataset to show its effectiveness.

**Strengths:**

1. The use of prototypical clustering for rPPG signals is a novel approach that effectively captures critical physiological signals.
2. The method outperforms existing models in terms of both accuracy and computational efficiency.

**Limitations:**

1. The initialization process for prototypes in this study involves using an MSTmap, which is an aggregation of average pixel values from each facial region of interest (ROI) over time. While this method simplifies the input data by reducing its dimensionality, it also raises concerns about the potential loss of critical spatial information. Averaging pixel values can smooth out fine details that might be crucial for detecting subtle physiological changes.

2. The “learn-and-cluster” strategy mentioned in the paper appears to draw inspiration from the deep clustering technique described by Caron et al. (2018). How about directly apply deepcluster on this task? This could be a baseline.
 [1]Caron, Mathilde, et al. "Deep clustering for unsupervised learning of visual features." Proceedings of the European conference on computer vision (ECCV). 2018.

3. The paper could benefit from a more detailed discussion on scenarios where Cluster-Phys underperforms or fails, providing insights into potential limitations or areas for improvement.  For example, how it performs when landmark is incorrectlty detected thus ROI is wrongly determined.

**Suitability:**

2

---

### Meta-Review · Area_Chair_vWbD · 2024-07-03

**Recommendation:** Accept (Oral)
**Confidence:** 5

**Metareview:**

All reviewers agree that the proposed method is interesting and innovative. All reviewers comment that the facial ROI Prototypical Clustering method to progressively learn and utilize representative ROI prototypes is novel and contribute significantly to the increased performance of the presented approach. The reviewers are satisfied with the presented experimental study. The rebuttal addressed a large majority of additionally raised questions. The authors are advised to include the relevant explanation in the final version of the paper. After reading the rebuttal, two of the reviewers upgraded the final score. Given a general appreciation of the work by the reviewers, I believe that the paper will be of interest to the audience attending ACM MM and would recommend a presentation of the work as a poster.